# Metabolomic Profiling Reveals the Anti-Herbivore Mechanisms of Rice (*Oryza sativa*)

**DOI:** 10.3390/ijms25115946

**Published:** 2024-05-29

**Authors:** Chengzhen Gu, Yujia Zhang, Mengmeng Wang, Yangzheng Lin, Bixue Zeng, Xinyu Zheng, Yuanyuan Song, Rensen Zeng

**Affiliations:** 1Key Laboratory of Ministry of Education for Genetics, Breeding and Multiple Utilization of Crops, College of Life Sciences, Fujian Agriculture and Forestry University, Fuzhou 350002, China; guchengzhen1984@126.com (C.G.); zhangyujia0113@163.com (Y.Z.); wangmengmeng2404@163.com (M.W.); 18359164616@163.com (Y.L.); zengbixue1997@163.com (B.Z.); zxy0621@fafu.edu.cn (X.Z.); 2Key Laboratory of Ministry of Education for Genetics, Breeding and Multiple Utilization of Crops, College of Agriculture, Fujian Agriculture and Forestry University, Fuzhou 350002, China

**Keywords:** *Oryza sativa*, anti-herbivore, induce, anti-insect activity

## Abstract

The use of secondary metabolites of rice to control pests has become a research hotspot, but little is known about the mechanism of rice self-resistance. In this study, metabolomics analysis was performed on two groups of rice (T1, with insect pests; T2, without pests), indicating that fatty acids, alkaloids, and phenolic acids were significantly up-regulated in T1. The up-regulated metabolites (*p*-value < 0.1) were enriched in linoleic acid metabolism, terpene, piperidine, and pyridine alkaloid biosynthesis, α-linolenic acid metabolism, and tryptophan metabolism. Six significantly up-regulated differential metabolites in T1 were screened out: *N*-*trans*-feruloyl-3-methoxytyramine (**1**), *N*-*trans*-feruloyltyramine (**2**), N-*trans*-*p*-coumaroyltyramine (**3**), *N*-*cis*-feruloyltyramine (**4**), *N*-phenylacetyl-L-glutamine (**5**), and benzamide (**6**). The insect growth inhibitory activities of these six different metabolites were determined, and the results show that compound **1** had the highest activity, which significantly inhibited the growth of *Chilo suppressalis* by 59.63%. Compounds **2**–**4** also showed a good inhibitory effect on the growth of *Chilo suppressalis*, while the other compounds had no significant effect. RNA-seq analyses showed that larval exposure to compound **1** up-regulated the genes that were significantly enriched in ribosome biogenesis in eukaryotes, the cell cycle, ribosomes, and other pathways. The down-regulated genes were significantly enriched in metabolic pathways, oxidative phosphorylation, the citrate cycle (TCA cycle), and other pathways. Eighteen up-regulated genes and fifteen down-regulated genes from the above significantly enriched pathways were screened out and verified by real-time quantitative PCR. The activities of detoxification enzymes (glutathione S-transferase (GST); UDP-glucuronosyltransferase (UGT); and carboxylesterase (CarE)) under larval exposure to compound **1** were measured, which indicated that the activity of GST was significantly inhibited by compound **1**, while the activities of the UGT and CarE enzymes did not significantly change. As determined by UPLC-MS, the contents of compound **1** in the T1 and T2 groups were 8.55 ng/g and 0.53 ng/g, respectively, which indicated that pest insects significantly induced the synthesis of compound **1**. Compound **1** may enhance rice insect resistance by inhibiting the detoxification enzyme activity and metabolism of *Chilo suppressalis*, as well as promoting cell proliferation to affect its normal growth and development process. The chemical–ecological mechanism of the insect resistance of rice is preliminarily clarified in this paper.

## 1. Introduction

Due to their sequestered growth, plants develop a range of defenses when being eaten by insects, including constitutive and induced defenses. The induced defense of plants can only be manifested in the presence of exogenous stimuli. For example, plants can produce various induced defense responses after being infested by phytophagous insects and then develop insect resistance through changes in physiology, biochemistry, and morphological characteristics [1].

The induced defense of plants against insects is mainly dependent on the secondary metabolites of plants. Secondary metabolites are the main chemical weapons of plants’ defense that are formed by plants through physiological and biochemical mechanisms in the long-term evolutionary process to cope with a variety of environmental stresses, constituting an important part of the plants’ defense and anti-stress systems [2]. A variety of toxic secondary metabolites, such as steroidal compounds, terpenes, phenols, indoles, and nitrogenous chemicals are produced when plants are subjected to insect pests. These substances can resist, avoid, poison, hinder the growth and development of insects, or attract natural enemies [3,4]. Insects may show a significant decrease in their pupation rate and survival rate, as well as a decrease in food intake when they are directly exposed to or fed with secondary metabolites having anthelmintic activity. These defensive secondary metabolites are normally present at relatively low levels in plants and are not essential, but they are important for plants’ role in resisting environmental stresses and responding to external stimuli, especially against insect damage [5]. For example, benzoxazinoids (BXs) are important defensive secondary metabolites in plants that are both anti-insect and anti-disease, and the BXs glucosides in the vesicles are interpreted hydrologically to release toxic glycosidic ligands for defense against insects when plants are eaten by herbivorous insects [6]. The concentration of the toxin mexofuranoside is very low in *Pastinaca sativa* roots and rapidly rises when the plants are damaged [7]. When *Spodoptera litura* feeds on tobacco leaves (*Nicotiana tabacum*), the nicotine content in the leaves increases for defensive purposes [8,9]. 3-deoxyanthocyanidins in wild-type *Sorghum bicolor* (L.) Moench significantly increased the mortality of maize aphids (*Rhopalosiphum maidis* Fitch) compared with ineffective mutants lacking them. The content of gamma-aminobutyric acid (GABA) increases when *Choristoneura rosaceana* crawls on soybean leaves, which inhibits the neurotransmission of herbivorous insects, reduces their development rate, and promotes their death [10]. Insects can feed on solid plants by stinging or chewing, and the plants activate their own defense mechanism under the action of inducing factors in the insects’ saliva, producing changes in proteins and enzymes and ultimately inducing the synthesis of defensive secondary metabolites in the metabolic network of the energy and materials in the plants in response to biotic stresses [1]. Therefore, it is possible to study the chemical composition of plants after feeding by insects and to explore the basis of substances with anti-insect activity.

Rice (*Oryza sative* L.) is an important food crop and is cultivated over a huge area across the world [11]. Insect pests are one of the main causes of rice yield reduction, and the economic losses caused by insect pests in the world are close to USD 10 billion every year [12]. The use of chemical pesticides poses a potential threat to food safety and destabilizes agroecosystems [13]. Therefore, the use of rice resistance is an important method of modern crop pest control.

When rice is eaten by herbivorous insects, it is induced to produce and accumulate a large number of targeted defensive substances, which affect a series of physiological and biochemical processes, such as molting, reproduction, metabolism, and energy conversion, in the insects [14]. The secondary metabolites produced by pest stress can be transferred from affected parts to the whole rice plant, and most of these metabolites play a regulatory role in the growth and development of herbivorous insects [15]. Tryptamine levels in rice leaves were 12 times higher than those in the control group when eaten by *Chilo suppressalis*, and the receptivity of leaf tissue to larvae was negatively correlated with the level of tryptamine on the leaf surface [16,17]. The secondary metabolite tryptamine is involved in plant defense. In addition, 2 terpenoids and 11 unknown compounds were identified in rice stressed by brown plant hopper, rice leaf roller, and *Chilo suppressalis* [18]. But, so far, the material basis for insect resistance activity in rice remains unexplained.

In this paper, we analyzed two different treatments of rice (insect-infested (T1) and non-infested (T2)) by metabolomics, screened out unique metabolites in T1, and evaluated the insect-resistant activities of different metabolites. An analytical method was established for the determination of anti-insect active substances, and their contents were found in T1 and T2. Detoxification enzyme activity and transcriptome sequencing were performed to reveal the anti-insect mechanisms of these substances. The results of this study could elucidate the chemical–ecological mechanisms of insect resistance in rice, promote the sustainable development of agriculture, and guarantee food security.

## 2. Results

### 2.1. Results of Metabolites

Metabolomics tests were performed on T1 and T2, and the quality control results showed that the proportion of substances with CV values of less than 0.5 in the T1 and T2 samples was higher than 85% (Figure 1A), indicating that the overall experimental data were stable. Repeated correlation assessment (Figure 1B) showed good reproducibility within the sample group and large differences between the groups. The PCA score plots of the T1 and T2 samples (Figure 1C) showed that the cumulative interpretation rate of the first two principal components was 75.3%. PC1 (61.3%) and PC2 (14%) could basically reflect the main characteristic information of all samples. The separation trend of the two groups of samples was obvious, indicating that the metabolite composition of the rice induced by pests had obviously changed. The prediction parameters of the OPLS-DA model (Figure 1D) were Q^2^ = 0.981 (*p* < 0.005) and R^2^Y = 1 (*p* < 0.005), indicating the best mode.

Metabolites with statistically significant differences were screened based on OPLS-DA analysis, with fold change ≥ 2, fold change ≤ 0.5, and VIP ≥ 1 as conditions. As shown in the volcano plot (Figure 2A), 398 differential metabolites were obtained. Compared with the T2 group, 354 metabolites in T1 were significantly up-regulated, 44 compounds were significantly down-regulated, and the overall metabolite content showed an up-regulated trend. The cluster analysis of differential metabolites (Figure 2B) showed that lipids (28.64%), alkaloids (13.57%), and phenolic acids (17.84%) accounted for a large proportion of the differential metabolites. The content of flavonoids was mostly down-regulated, and a small part was up-regulated. Amino acids and their derivatives, organic acids, nucleotides and their derivatives, lignans, coumarins, and terpenes, with relatively low proportions, were mostly up-regulated.

The KEGG enrichment diagram (Figure 3A) shows that 398 different metabolites were enriched in 79 metabolic pathways, and the enrichment degree was greater when the Rich factor was higher, and the lower the *p*-value, the more significant the enrichment. Among them, 14 metabolic pathways were highly enriched (*p*-value < 0.5). These included linoleic acid metabolism, terpene, piperidine, and pyridine alkaloid biosynthesis, α-linolenic acid metabolism, and so on. Among them, the top 20 metabolic pathways with the highest abundance tended toward a score of 1 in combination with the analysis of the differential abundance score (Figure 3B), indicating that the expressions of differential metabolites in T1 show an overall up-regulated trend compared with T2 (the more the DA score tends to be 1, the more the overall expression of the pathway tends to be up-regulated; the more it moves to −1, the lower it is). It was found that metabolic pathways with significantly up-regulated metabolites (*p*-value < 0.1) were linoleic acid metabolism, terpene, piperidine, and pyridine alkaloid biosynthesis, α-linolenic acid metabolism, lysine degradation, tryptophan metabolism, and phenylalanine metabolism, which may be important metabolic pathways for insect resistance in rice.

After feeding, 71 kinds of alkaloid metabolites in rice were significantly changed (Figure 4), among which 70 metabolites were up-regulated and 1 metabolite was down-regulated. Phenolamines were the most abundant among them including 29 kinds, which suggests that phenolamines may be involved in the resistance of rice to these insects. Finally, we screened six phenolamines that were significantly up-regulated differential metabolites—*N*-*trans*-feruloyl-3-methoxytyramine (**1**), *N*-*trans*-feruloyltyramine (**2**), N-*trans*-*p*-coumaroyltyramine (**3**), *N*-*cis*-feruloyltyramine (**4**), *N*-phenylacetyl-L-glutamine (**5**), and benzamide (**6**)—and anti-insect activities of these six compounds (Figure 5) were determined.

### 2.2. Results of Insecticidal Activity

The results show that at a concentration of 100 μg/g, compound **1** (Figure 6) could significantly inhibit the growth of *Chilo suppressalis* at 3, 6, and 9 days, and the inhibitory rate was the highest in the first 3 days, reaching 59.63%, and then decreased and stabilized at 42%. At a concentration of 100 μg/g, compounds **2**–**4** showed better anti-insect activity. Compounds **5**–**6** had no significant effect on the growth of *Chilo suppressalis*.

### 2.3. Contents of Compound ***1*** in Different Treatments of Rice

The concentration and peak area of compound **1** were used as the horizontal coordinate (X) and the vertical coordinate (Y), respectively. The standard curve of compound **1** was drawn (Figure 7), and the regression equation was obtained.
$$Y=45404.2X+40.1898\;r^{2}=0.999$$

The contents of compound **1** in T1 and T2 were 8.55 ng/g and 0.53 ng/g, respectively. As shown in Figure 8, the content of compound **1** in T1 was significantly higher than that in T2, indicating the pest-induced synthesis of compound **1**, and improved the resistance of rice to *Chilo suppressalis*.

### 2.4. Effects of Compound ***1*** on Detoxification Enzyme Activities

The enzyme activity assay showed that larval exposure to compound **1** significantly reduced the activities of GST, and the inhibition ratio was 41% (Figure 9A). The UGT (Figure 9B) and CarE (Figure 9C) activities showed no significant differences between the RT and CK groups.

### 2.5. Results of Transcriptome Sequencing

#### 2.5.1. Quality of Transcriptome Sequencing

*Chilo suppressalis* was fed an artificial diet containing 100 μg/g of compound **1** for 9 days as the treatment group (RT) and an artificial diet containing an equal volume of DMSO for 9 days as the control group (CK). RNA-Seq analysis was performed for the two groups. The raw reads were 22633197, 23337632, 22768375, 24100758, 23487995, and 21570901, respectively. The clean bases after filtering low-quality sequences were 21769138, 22542267, 21925915, 23138018, 22675237, and 20795171. The subsequent analysis was based on clean bases for high-quality data analysis.

The error rate of the clean data obtained from the quality summary of the sequencing data (Table 1) was 0.03%, which met the standard. The bases with Phred values greater than 20 and 30 accounted for 97.34–97.63% and 92.3–92.91% of the total bases, respectively, indicating that the data quality was strictly controlled, the overall quality was good, and the GC content was about 42%, which belonged to the normal range.

#### 2.5.2. Quality of Transcripts

As clean reads need to be spliced for transcriptomes without reference genomes, three independent software modules of Trinity (v2.6.6), namely, Inchworm, Chrysalis, and Butterfly, were used, therefore, to splice all clean reads and obtain preliminary transcripts. Then, redundant transcripts were aggregated by the Corset hierarchical clustering of the transcripts, so that the transcripts with expression differences between the samples were separated from the original cluster, and new clusters were established. Each cluster was defined as “Gene”. The stitching quality was evaluated by using BUSCOs (benchmarking universal single-copy orthologs), with high accuracy and integrity, as shown in Figure 10.

#### 2.5.3. Analysis of Differentially Expressed Genes between CK Group and RT Group

The results of the repeated correlation assessment between the CK group and RT group (Figure 11A) showed good reproducibility within the sample group and large differences between the groups. Differentially expressed genes (DEGs) were screened with padj < 0.05 and |log_2_(foldchange)| > 1 as conditions. The DEGs had statistically significant expression levels. The results are shown in the volcano plot (Figure 11B). A total of 5219 DEG were detected, among which 3623 genes in the RT group were up-regulated (red dots), 1596 genes were down-regulated (green dots), and 42,657 genes were not significantly co-expressed (blue dots) compared with the CK group. The DEG expression showed an overall up-regulated trend.

#### 2.5.4. Functional Annotation and Enrichment Analysis of DEGs

Goseq was used for the GO enrichment analysis. The GO database divides gene functions into three categories: biological process (BP), cellular component (CC), or molecular function (MF). The significant enrichment analysis with the GO function (padj < 0.05) showed that biological functions were significantly correlated with the DEGs. The results show (Figure 12A) that the differentially expressed genes of the CK group and the RT group were annotated as MF, BP, and CC, and more of them were annotated as BP and MF.

In the RT group, the up-regulated DEGs were significantly enriched in the cellular nitrogen compound metabolic process, ATPase activity, biosynthetic process, intracellular interaction, organelle, chromosome, and phosphatase activity terms (Figure 12B). The down-regulated DEGs in the RT group were only significantly enriched in the oxidoreductase activity term related to energy metabolism (Figure 12C).

According to the KEGG pathway analysis, the significantly up-regulated genes were mainly enriched in the ribosome, ribosome biogenesis in eukaryotes, cell cycle, DNA replication, proteasome, mismatch repair, nucleotide excision repair, aminoacyl-tRNA biosynthesis, and other pathways (Figure 13A). All these pathways are involved in cell cycle regulation and cell proliferation.

The significantly down-regulated genes were mainly enriched in metabolic pathways, oxidative phosphorylation, the citrate cycle (TCA cycle), thermogenesis, carbon metabolism, pyruvate metabolism, the biosynthesis of secondary metabolites, drug metabolism—cytochrome P450, and other pathways (Figure 13B). These pathways are mainly related to energy metabolism.

### 2.6. Verification by RT-qPCR

Thirty-three differentially expressed genes were randomly selected and verified by RT-qPCR, including eighteen up-regulated and fifteen down-regulated differential genes. The 18 up-regulated differentially expressed genes belonged to the biosynthetic, cell cycle, DNA replication, and metabolic pathways of eukaryotic ribosomes. The 15 down-regulated differential genes belonged to metabolic pathways, oxidative phosphorylation, and the tricarboxylic acid cycle metabolism pathway. In the significantly up-regulated genes (Figure 14A), only REXO1 and MCM3 could be detected in the CK group, while the other target genes were only detected in the RT group, and these genes were significantly up-regulated in the RT group. In the down-regulated genes (Figure 14B), the gene expression levels in the RT group were significantly lower than those in the CK group, which was consistent with the transcriptome sequencing results. The up-regulated genes were significantly enriched in metabolic pathways related to cell proliferation, and the down-regulated genes were significantly enriched in the pathways related to material and energy metabolism. The results confirm that compound **1** disrupted the normal growth and development of *Chilo suppressalis* larvae by promoting abnormal cell proliferation, inhibiting metabolism and energy supply, and inhibiting the growth of the larvae.

## 3. Discussion

The research on rice resistance has been ongoing for a long time, and the research on rice resistance to insects has been a research hotspot recently [19,20]. The mechanism of rice resistance to insects is mostly explained at the molecular level, and a number of insect resistance genes have been obtained from rice, but the chemical basis of insect resistance in rice has still not been clearly explained, resulting in the functions of many resistance genes not being verified [21,22,23].

In this paper, the metabolomics analysis of the T1 and T2 groups showed that the metabolic pathways with significantly up-regulated metabolites (*p*-value < 0.1) were linoleic acid metabolism, terpene, piperidine, and pyridine alkaloid biosynthesis, α-linolenic acid metabolism, and tryptophan metabolism. Studies have shown that the accumulation of plant fatty acids and their derivatives can effectively improve plant insect resistance, and especially the content of fatty acids with C18 chains can affect the signaling pathway mediated by salicylic acid and improve plant resistance to insect pests [24,25]. Fifteen fatty acids with C18 chains were enriched in the linoleic acid metabolic pathway and were all up-regulated, suggesting that there might be similar mechanisms of insect resistance in rice. The α-linolenic acid metabolic pathway was enriched with nine metabolites: jasmonic acid, 13S-hydroxy-9Z,11E,15Z-octadecatrienoic acid, 9-hydroxy-12-oxo-15(Z)-octadecenoic acid, 13(s)-hydroperoxy-(9z,11e,15z)-octadecatrienoic acid, methyl jasmonate, 9-hydroxy-10,12,15-octadecatrienoic acid, 2R-hydroxy-9Z,12Z,15Z-octadecatrienoic acid, 17-hydroxylinolenic acid, 9-hydroxy-12-oxo-10(E), and 15(Z)-octadecadienoic acid. The contents of these metabolites were significantly increased in group T1. The increased contents of jasmonic acid and its derivative methyl jasmonate indicated that the signal transduction pathway of JA synthesis was activated [26], and JA was the main defense hormone to activate plant resistance to herbivorous insects, suggesting that JA is also involved in regulating rice resistance to insects. Studies [27] have shown that the pathway of tryptophan metabolism can affect the defense response of rice against brown planthoppers. In this pathway, the OsTrp1 gene negatively regulates the salicylic acid (SA) pathway of rice. Deletion of the OsTrp1 gene reduces the survival rate of nymphs and improves insect resistance.

The contents of alkaloids in rice induced by insect infestation were significantly up-regulated. Alkaloids are nitrogen-containing organic compounds that do not play a major role in plant growth and development, but many are toxic to herbivores. Alkaloids can produce deterring, toxic, and anti-feeding effects on herbivorous insects; change the physiological and biochemical processes in insects; and inhibit or disturb their metabolism, growth, and development [28,29]. Alkaloids can inhibit the synthetic repair of DNA as well as transcription and translation processes; can inhibit the biosynthesis of resistant proteins in insects, causing toxicity to them; and can also affect insects’ nervous systems from the receptors to the neurotransmitters [28]. Phenolamine compounds **1**–**4**, belonging to alkaloids, had significant anti-insect activity, which indicates that phenolamines have antifeedant or gastrotoxic activity against the insect. The anti-insect activity decreased from compounds **1** to **4** at the same concentration. Regarding their structures, the double bond in compounds **1**–**3** is *trans*, while in compound **4**, it is *cis*, which suggests that the anti-insect activity of trans double bonds is better. Previous studies have shown that exogenous stress and insect harm can induce the synthesis of phenolamines. The exogenous application of 4-chlorophenoxyacetic acid can induce a chemical defense to protect rice plants from white-backed planthoppers. Meanwhile, 4-chlorophenoxyacetic acid increases the levels of phenolamines and flavonoids [30]. The contents of phenolamines and phenolic acids in rice increased when the rice was damaged by brown planthoppers [31]. The content of phenolamines in compound **1** increased by eight times when the rice was infested by pests.

The results of the detoxification enzyme activity related to insect metabolic resistance show that compound **1** had no significant effect on CarE and UGT activities but significantly inhibited GST activity, which was 0.59 times higher in the treatment group than in the control group. Studies have shown that GST is mainly involved in the second stage of pesticide metabolism, which can transport and dissolve the hydrolyzed products after a series of biochemical reactions in the first stage and catalyze the combination of reduced glutathione (GSH) with electrophilic toxic substances, increasing its water solubility, which is conducive to excretion. Because it also has peroxidase activity, it can play an important role in preventing oxidative stress caused by the ingestion of toxic substances [32]. By genome and transcriptome sequencing, 23 CsGSTs genes were identified in abamectin-treated *Chilo suppressalis*, and when the GSTs were inhibited, the susceptibility of the larvae to abamectin was increased, and the survival rate of the larvae was decreased [33]. Compound **1** may reduce the detoxification ability and inhibit larval weight gain by negatively regulating GSTs in a similar manner.

## 4. Conclusions

When infested by pests, rice resists insects by immediately up-regulating fatty acids, alkaloids, and phenolic acids. These metabolites were enriched into the pathways of linoleic acid metabolism, terpene, piperidine, and pyridine alkaloid biosynthesis, α-linolenic acid metabolism, and tryptophan metabolism. Phenolamine compounds **1**–**4** showed better anti-insect activity according to the verification results.

Compound **1** may reduce the detoxification ability of *Chilo suppressalis* against exogenous drugs by inhibiting the activity of the insect detoxification enzyme GST. Meanwhile, compound **1** reduces the activity of larvae by promoting the abnormal proliferation of larval cells, inhibiting metabolism and the energy supply, disrupting the normal growth and development process, and making the growth and development process tend to shorten the larval stage and accelerate emergence. Finally, it can inhibit the growth and development of *Chilo suppressalis*.

## 5. Materials and Methods

### 5.1. Insects and Rice

*Oryza sativa* L. spp. *indica*, cv. Oochikara was provided by the Laboratory of Crop Resistance and Chemical Ecology, the Fujian Agriculture and Forestry University. *Chilo suppressalis* Walker egg mass was purchased from Hubei shengnong technology and cultured in a climate chamber.

### 5.2. Reagents and Instruments

Standard samples of compounds **1**–**6** were purchased from Chengdu aierfa company. The instruments used were electronic scales (Quintix224-1cn, Sartorius, Göttingen, Germany), a freeze dryer (Scientz-100F, Hangzhou City, China), an ultra-high-performance liquid chromatograph (Nexera X2, SHIMADZU, Kyoto City, Japan), an MS/MS (4500 QTRAP, Applied Biosystems, Waltham, MA, USA) Vortex-Genie 2 Vortex oscillator (G560E, Scientific Industries, New York, NY, USA), and an ultra-high-performance liquid chromatography–tandem triple quadrupole mass spectrometer (Acquity UPLC/XEVO TQS MS, Waters, Milford, MA, USA).

### 5.3. Metabolites

#### 5.3.1. Extraction and Preparation of Samples

The rice seedlings were cultured with a Kimura B nutrient solution, changing the nutrient solution every three days, for 4 weeks. Third-instar *Chilo suppressalis* larvae were selected and starved for 24 h and were placed on the rice stems with a brush as the treatment group. Rice stems without insect damage were used as the control group. After 48 h, 2 cm above and below the insect sites of the rice stalks were cut off by scissors at room temperature, immediately put into liquid nitrogen, and stored at −80 °C. Two groups of rice materials were obtained: T1 (with insect damage) and T2 (without insect damage).

The samples of T1 and T2 were freeze-dried by a vacuum freeze-dryer (Scientz-100F). The freeze-dried samples were crushed using a mixer mill (MM 400, Retsch, Beijing, China) with a zirconia bead for 1.5 min at 30 Hz. An amount of 100 mg of each lyophilized powder was dissolved in 1.2 mL of a 70% methanol solution, vortexed for 30 s every 30 min, 6 times in total, and the samples were placed in a refrigerator at 4 °C overnight. Following centrifugation at 12,000 rpm for 10 min, the extracts were filtrated before UPLC-MS/MS analysis.

#### 5.3.2. UPLC and MS Conditions, Qualitative and Quantitative of Metabolites, and Data Processing and Analysis (See Appendix A)

### 5.4. Insecticidal Activity of Metabolites

Ten significantly up-regulated metabolites were selected from the T1 group by metabolomics analysis. The metabolites were dissolved in dimethyl sulfoxide (DMSO), mixed with an artificial feed at 50 °C in a certain proportion, and stirred evenly to obtain 100 μg/g of mixed toxic–artificial feed as the treatment group. The artificial feed with an equal volume of DMSO was used as the control group and stored in a refrigerator at 4 °C after coagulation. The third instars of *Chilo suppressalis* with a body weight of about 5 mg were chosen and starved for 3 h. The larvae were fed with the artificial feed, and five biological replicates were set. The weight of each *Chilo suppressalis* specimen was recorded at three, six, and nine days, respectively, and they were frozen on the last day.

### 5.5. Determination of Compound ***1*** in Rice

#### 5.5.1. Sample and Standard Preparation

The rice samples (T1 and T2) were ground to powders in liquid nitrogen and ultrasonically extracted with 70% methanol for 20 min. The supernatants were taken and filtered for testing.

Compound **1** was dissolved with methanol, and six standard solutions with different concentrations (0.001, 0.01, 0.05, 0.1, 1, and 10 ng/mL) were prepared.

#### 5.5.2. UPLC-QqQ MS Conditions

The UPLC conditions were as follows. Column: Waters BEH C18; 1.7 µm, 100 mm × 2.1 mm. Mobile phase A: pure water with 0.1% formic acid; mobile phase B: acetonitrile with 0.1% formic acid. Gradient: 5–100% B for 0–4 min; 100% B for 4–5 min; 100–5% B for 5–5.1 min; and 5% B for 5.1–8 min. The column temperature was 40 °C. The injection volume was 2 µL. The flow rate was 0.3 mL/min.

The MS conditions were as follows. The ESI ion source was in positive-ion mode. The cone hole voltage and the capillary voltage were 30 V and 1.0 kV, respectively. The gas flow rate of the cone hole was set to 150 L/h. The solvent removal temperature was 450 °C. The de-solvent flow rate was 900 L/h. The MRM mode was used for the scan.

The construction of the standard curve was as follows. Six standard solutions with different concentrations (0.001, 0.01, 0.05, 0.1, 1, and 10 ng/mL) were injected for analysis. With the concentration of the standard solution as the horizontal coordinate (X) and the peak area as the vertical coordinate (Y), the standard curve of compound **1** was drawn, and the regression equation was obtained, which was used to calculate the content of compound **1** in each test sample. 

The experimental data were analyzed by GraphPad Prism 8 software and IBM SPSS Statistics 25 software. One-way ANOVA was used to analyze the data. Student’s *t*-test was used to analyze differences between the two groups. When *p* < 0.05, it was considered that there was a significant difference between the two groups.

### 5.6. Detoxifying Enzyme Activity Assay of Chilo Suppressalis

#### 5.6.1. The Procedure for Aqueous Enzymatic Extraction in CarE and UGT Assays

Nine larvae with similar weights were selected from the RT and CK groups and randomly divided into three groups. The samples were ground to powders in liquid nitrogen and thoroughly ground in an ice bath by a grinder after adding 200 μL (0.04 M; pH 7.0) of a phosphoric acid buffer. Then, they were capped and mixed by an oscillator and, finally, centrifuged (4 °C; 12,000 rpm) for 20 min. The supernatants were taken as the enzyme liquids to be measured.

#### 5.6.2. The Procedure for Aqueous Enzymatic Extraction in P450 and GST Assays

A phosphoric acid buffer (0.1 M; pH 7.5) was prepared by mixing 9.5 mL of a buffer with 1 mM EDTA, DTT, PTU, PMSF, and 10% glycerol. The samples and 200 μL of the phosphoric acid buffer were added into tubes and ground in an ice bath by a grinder. Then, 300 μL of the phosphoric acid buffer was added, and the contents were mixed by an oscillator and, finally, centrifuged (4 °C; 12,000 rpm) for 20 min. The supernatants were absorbed as the enzyme liquids to be tested.

The enzymatic activities of glutathione S-transferase (GST), uridine diphosphate glucuronyltransferase (UGT), and carboxylesterase (CarE) in the *Chilo suppressalis* specimens, which were fed with an artificial diet containing 100 μg/g of compound **1**, were detected by a method previously reported in [34].

### 5.7. Transcriptome Sequencing

#### 5.7.1. RNA Extraction

*Chilo suppressalis* specimens were fed with an artificial diet containing 100 μg/g of compound **1** for 9 days as the treatment group (RT group) and artificial feed containing DMSO for 9 days as the control group (CK group). The insects were frozen with liquid nitrogen and stored at −80 °C in an ultra-low-temperature refrigerator for further use.

Nine insects with similar weights were selected from the RT and CK groups and randomly divided into three groups. The total RNA from each of the 6 groups (RT and CK group × three replicates) was extracted with an Ultrapure RNA Kit (CWBIO, Beijing, China) according to the manufacturer’s instructions. Electrophoresis with 1.2% agarose gel was used to analyze the purity and integrity of the mRNA. The total RNA was used for RNA-Seq analysis.

#### 5.7.2. Construction of RNA-Seq Library, Quality Control, and Sequencing

Libraries were constructed using the NEBNext^®^ Ultra™ RNA Library Prep Kit for Illumina^®^ following the manufacturer’s instructions. The library quality was examined by an Agilent 2100 bioanalyzer. Sequencing was performed on the Illumina Novaseq 6000 platform by Novogene Co., Ltd. (Beijing, China).

#### 5.7.3. Differentially Expressed Genes’ Identification and Functional Enrichment Analysis

Differentially expressed genes (DEGs) were selected by the DESeq2 software (R4.1.2). Significant differences in gene expression were determined following the criteria of Log_2_fold change > 0.5 and an adjusted *p*-value (*P*adj) < 0.05. The obtained DEGs were then subjected to Gene Ontology (GO) and Kyoto Encyclopedia of Genes and Genomes (KEGG) pathway enrichment analyses.

### 5.8. Real-Time Quantitative PCR (RT-qPCR)

Via transcriptome comparative analysis, 18 up-regulated and 15 down-regulated differentially expressed genes were randomly selected and detected by fluorescence quantitative PCR with the method previously reported in [35]. (The primer of genes see Appendix A).

## Figures and Tables

**Figure 1 ijms-25-05946-f001:**
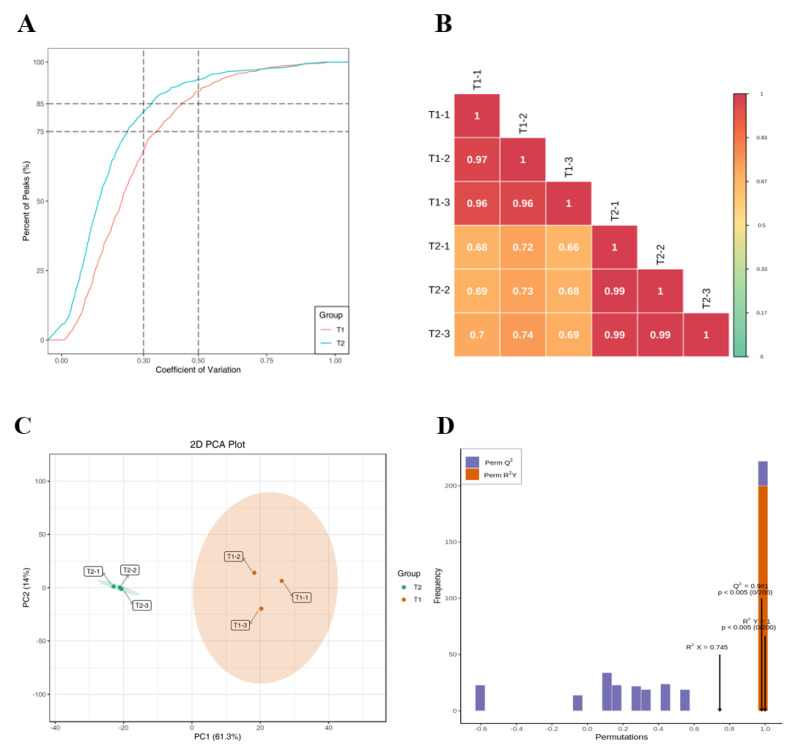
Qualitative, quantitative, and quality control analyses of metabolites. (**A**) Sample CV distribution map. (**B**) Heat map of correlation between samples. (**C**) PCA diagram of T1 and T2 samples. (**D**) OPLS–DA validation diagram.

**Figure 2 ijms-25-05946-f002:**
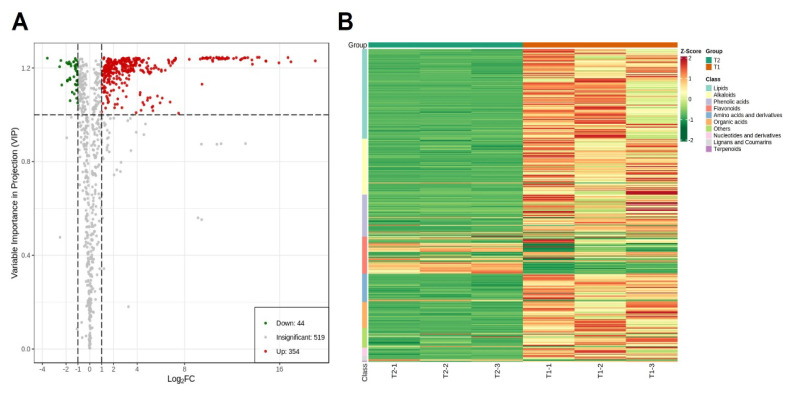
Volcano plot and clustering analysis of differential metabolites. (**A**) Volcano plot of differential metabolites between T1 and T2 groups. Red dots indicate an up-regulation of metabolite accumulation, while green dots indicate a down-regulation of metabolite accumulation. (**B**) Differential metabolite clustering heatmap between T1 and T2 groups. Z–score value displays a change in color from high (red) to low (green).

**Figure 3 ijms-25-05946-f003:**
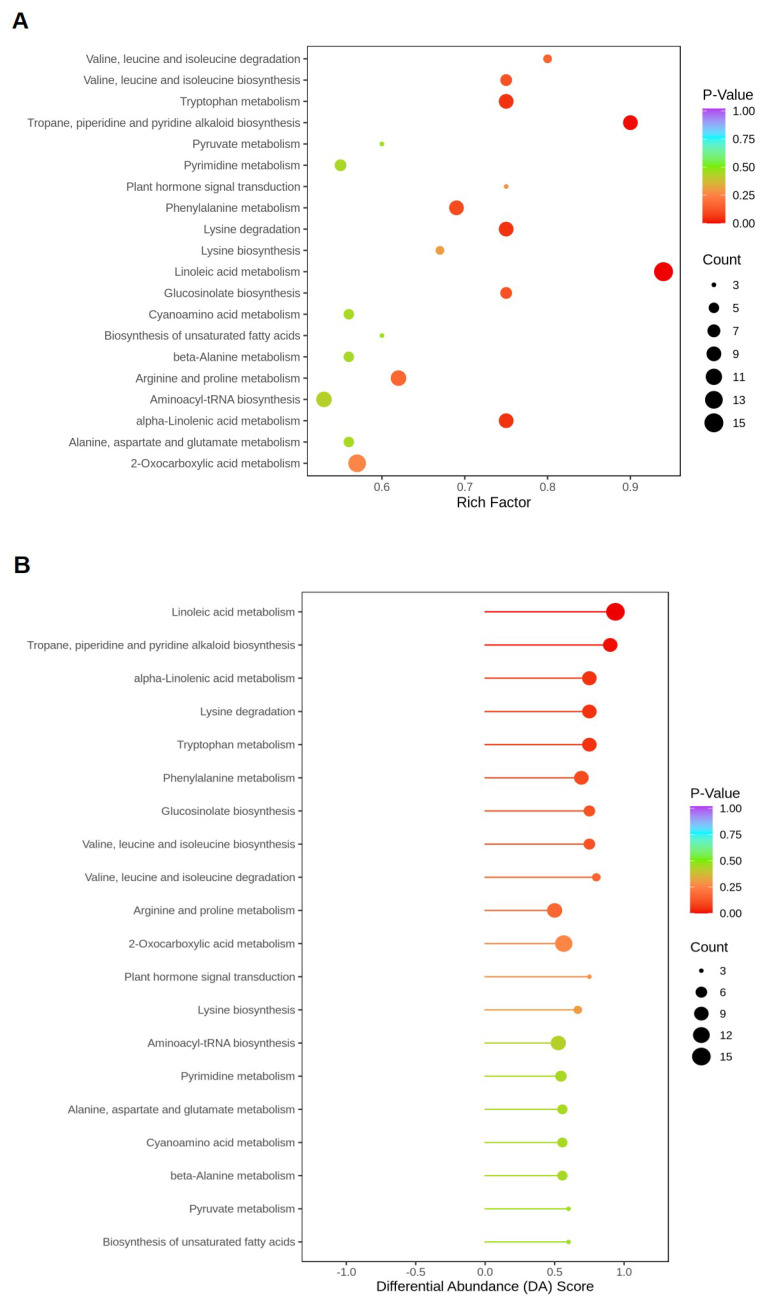
KEGG enrichment map and differential abundance score map of differential metabolites in T1 and T2 groups.

**Figure 4 ijms-25-05946-f004:**
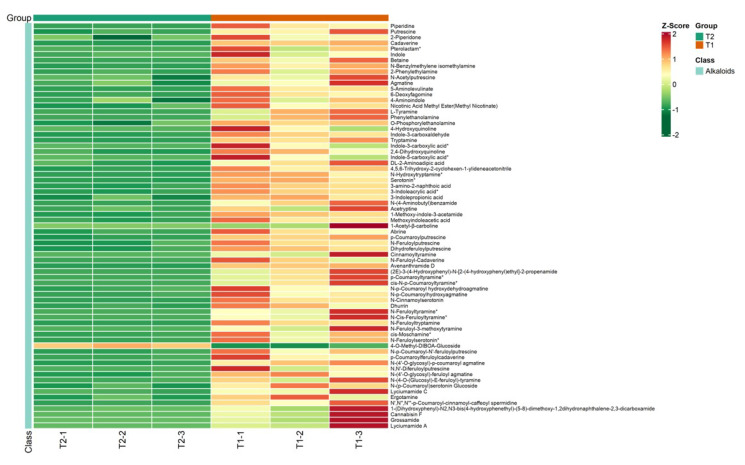
Cluster heat map of alkaloid differential metabolites in T1 and T2 (* represent the isomer).

**Figure 5 ijms-25-05946-f005:**
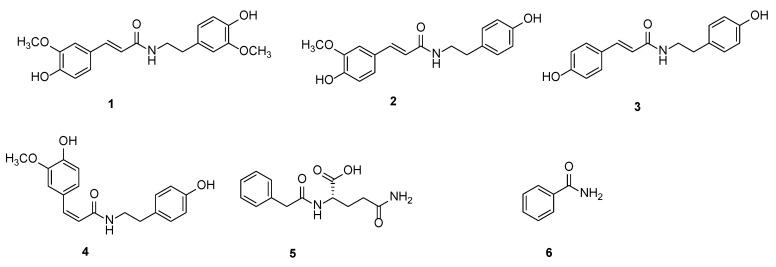
Compounds **1**–**6**.

**Figure 6 ijms-25-05946-f006:**
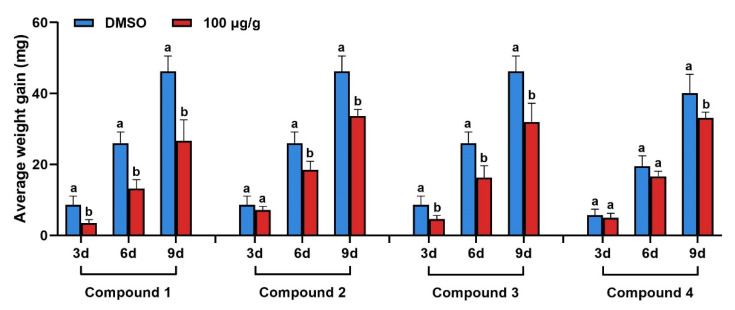
Effect of compounds **1**–**4** on the body weight of *Chilo suppressalis* and the change in body weight growth inhibition rate over time (a and b represent a significant difference between the two groups).

**Figure 7 ijms-25-05946-f007:**
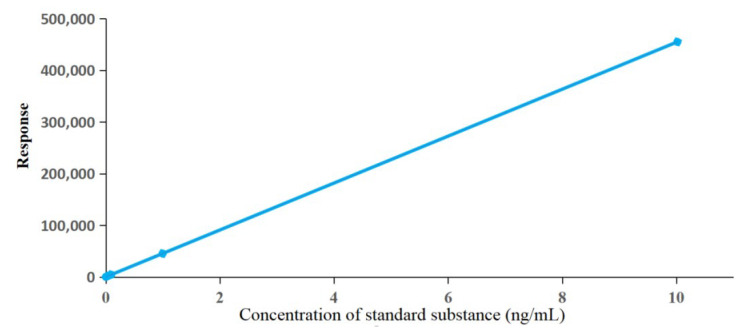
Standard curve of compound **1**.

**Figure 8 ijms-25-05946-f008:**
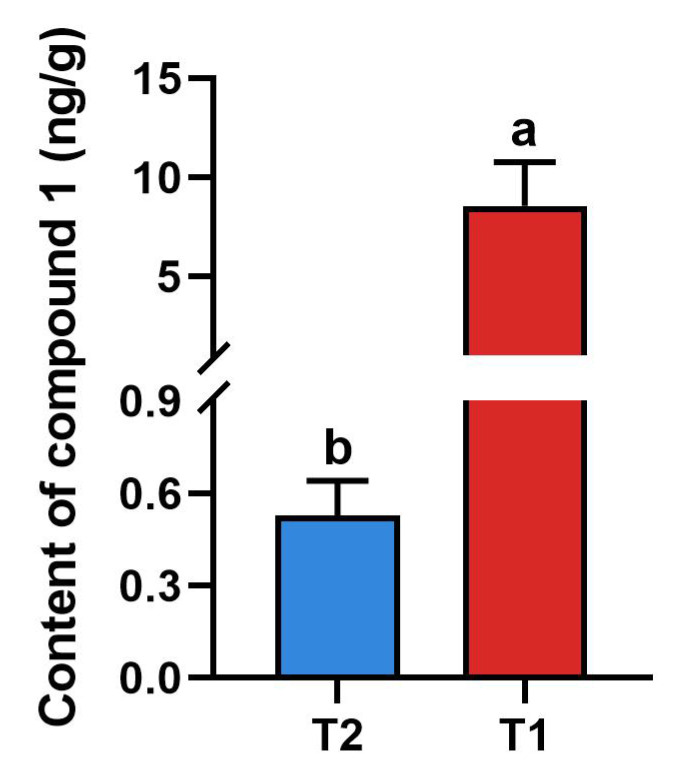
Absolute content of compound **1** in two groups of rice materials (a and b represent a significant difference between the two groups).

**Figure 9 ijms-25-05946-f009:**
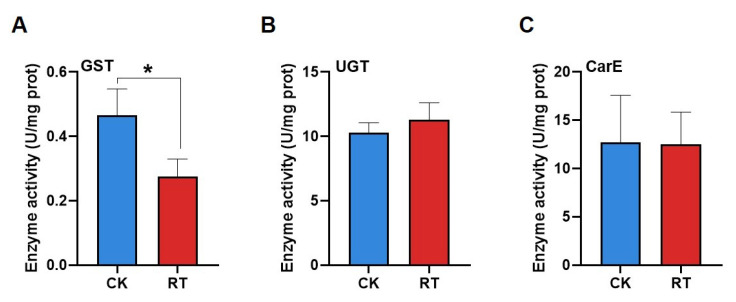
The effect of compound **1** on detoxifying enzymes’ activities in *Chilo suppressalis.* (**A**) GST enzyme activity, (**B**) UGT enzyme activity, and (**C**) CarE enzyme activity. Asterisks indicate significant differences (* *p* < 0.05; Student’s *t*-test) (Red represent the treatment group, blue represent the control group).

**Figure 10 ijms-25-05946-f010:**
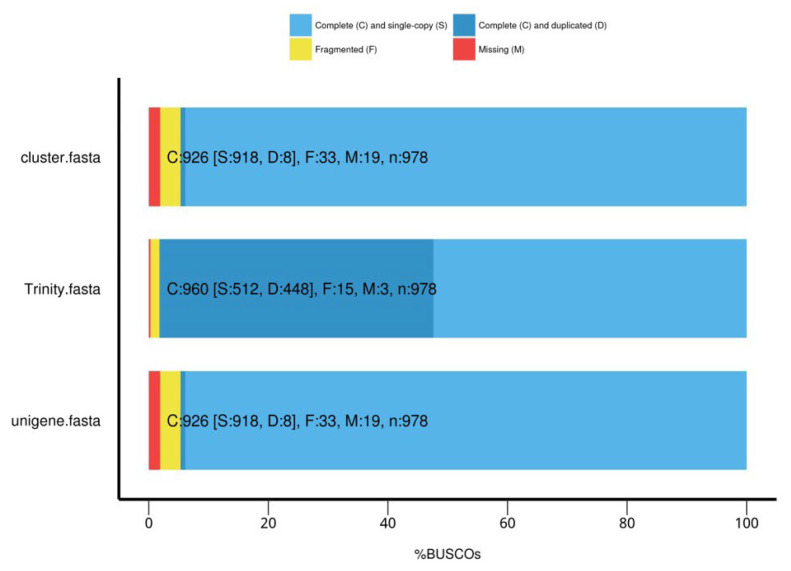
BUSCO assessment results.

**Figure 11 ijms-25-05946-f011:**
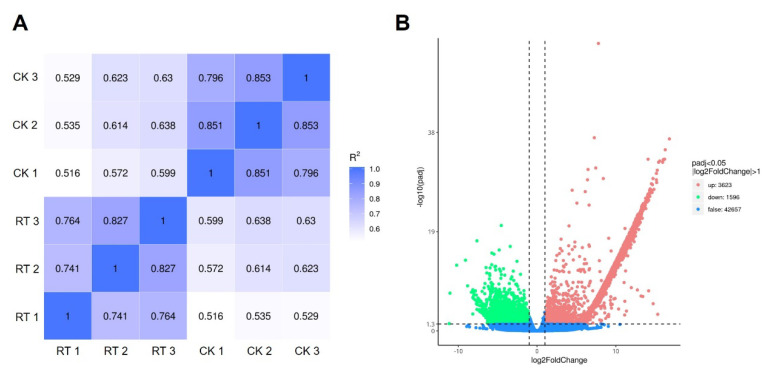
Correlation analysis and volcano plot of CK group and RT group.

**Figure 12 ijms-25-05946-f012:**
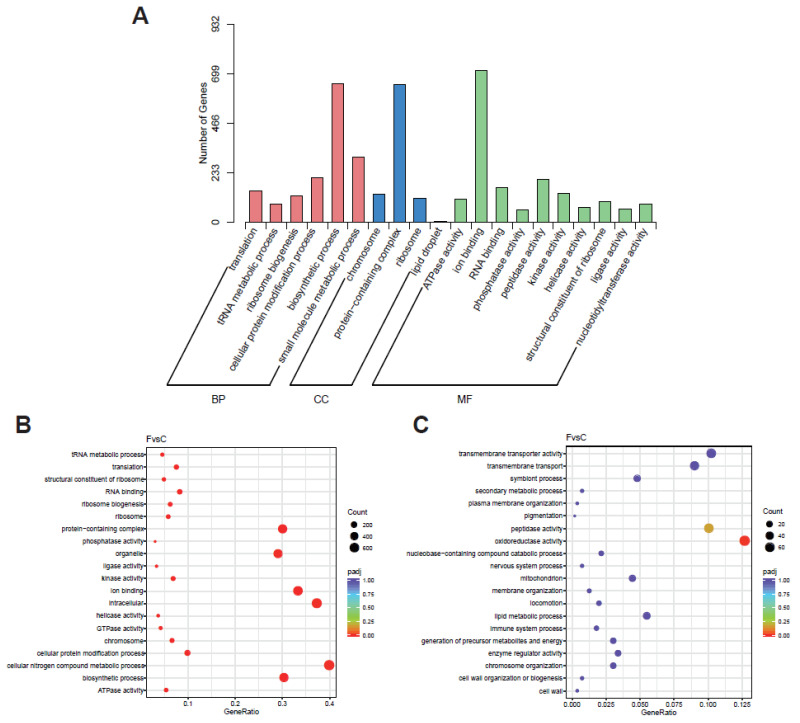
GO enrichment histogram and GO enrichment scatterplot of differential genes in CK and RT groups. (**A**) Enriched GO terms; (**B**) GO enrichment scatterplot of up–regulated genes in expression levels; and (**C**) GO enrichment scatterplot of down-regulated genes in expression levels.

**Figure 13 ijms-25-05946-f013:**
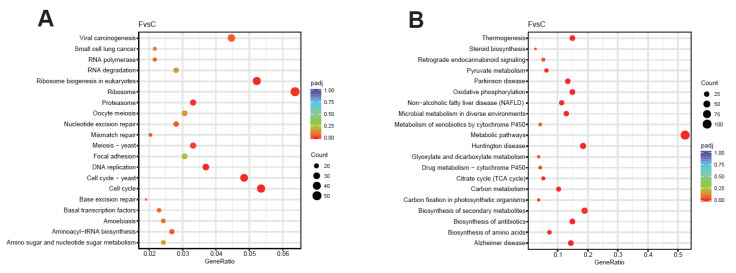
Scatterplots of KEGG enrichment in CK and RT groups. (**A**) KEGG enrichment scatterplot of up-regulated genes in expression levels; (**B**) KEGG enrichment scatterplot of down-regulated genes in expression levels.

**Figure 14 ijms-25-05946-f014:**
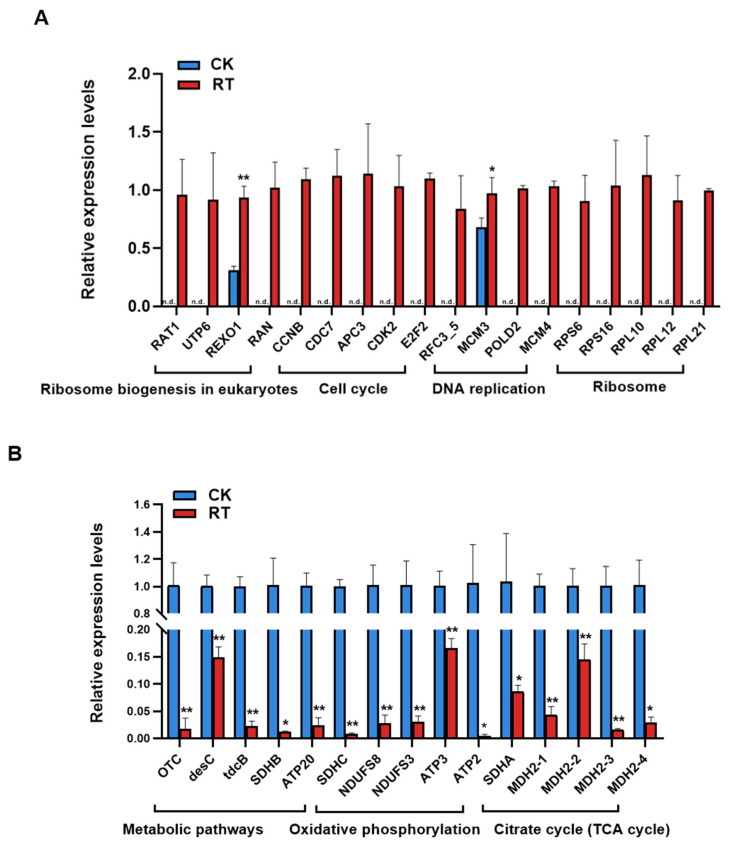
Effect of Compound **1** on gene expression of *Chilo suppressalis*. (**A**) The expression levels of up-regulated differentially expressed genes. (**B**) The expression levels of down-regulated differentially expressed genes (* and ** represent a significant difference between the two groups; n.d. represent the gene was not detected).

**Table 1 ijms-25-05946-t001:** Summary of sample sequencing data quality.

Sample	Raw Reads	Clean Reads	Clean Bases	Error Rate	Q20	Q30	GC_pct
CK1	22633197	21769138	6.5	0.03	97.47	92.59	41.64
CK2	23337632	22542267	6.8	0.03	97.34	92.3	41.98
CK3	22768375	21925915	6.6	0.03	97.56	92.71	41.00
RT1	24100758	23138018	6.9	0.03	97.37	92.48	43.40
RT2	23487995	22675237	6.8	0.03	97.41	92.48	43.41
RT3	21570901	20795171	6.2	0.03	97.63	92.91	42.88

## Data Availability

The original contributions presented in the study are included in the article/Appendix A, further inquiries can be directed to the corresponding author/s.

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
