# Peer review of "Metabolomic Profiling Reveals the Anti-Herbivore Mechanisms of Rice (Oryza sativa)"

_ijms, 2024, doi:10.3390/ijms25115946_

Round 1

Reviewer 1 Report

Comments and Suggestions for Authors

Comments on the Quality of English Language

There is minor grammar problems which must be considered. Refer to the comments

Author Response

Dear Prof. Sandra Wan

Thank you very much for your decision letter, concerning of our manuscript entitled “Metabolomic Profiling Reveals the Anti-herbivore Mechanism of Rice (Oryza sativa)” (ijms-2991558). According to the reviewers’ comments and suggestions, we have carefully revised our manuscript and our revisions are listed in the document designated as Response to Decision Letter. We provided the revised manuscript (ijms-2991558), the changes had been marked in red. Our revisions are listed as followings:

  • Responses to Reviewer 1:
  1. We had added citation in Line 82.
  2. We had added “The results of this study could elucidate……” in Line 103
  3. “.” was added after [23,24]
  4. Line 330 was revised
  5. Line 295 had been revised
  6. Company name was added in line 373
  7. “Rice stalks were cut off by scissors at room temperature” and added in line 388-389
  8. “it” was deleted in line 408
  9. Line 414-415, 480-481 had been revised followed comment
  10. “;” was replaced by “.”
  11. The concentration of the RNA was determined after extracted by Novogene Co. Ltd
  12. “Illumina Novaseq 6000 platform” was added in Line 463
  13. The primer of genes was supported in supporting information S2
  14. “.” was added in line 542 and Line 562. All the references were checked.

We hope that the revised manuscript will meet your magazine’s standard and is acceptable to be published in international journal of molecular sciences. Please feel free to contact me if you have any questions. We look forward to hearing from you regarding its disposition. Thank you again for your time and kind advice.

With best regards

Sincerely yours,

Cheng-Zhen Gu, Ph.D,

College of Life Science, Fujian Agriculture and Forestry University.

Reviewer 2 Report

Comments and Suggestions for Authors

This study provides valuable insights into the mechanism of rice self-resistance against insect pests, which is crucial for developing sustainable pest management strategies. Some minor issues need to be addressed especially in the discussion.

1. While the study provides valuable insights, it may only scratch the surface of the complex chemical-ecological mechanisms involved in rice self-resistance. Further research is needed to fully understand these mechanisms.

2. The study focuses on specific metabolites and their effects on a particular insect pest (Chilo suppressalis), which may limit the generalizability of the findings to other pests or plant species.

3. The use of bioactive compounds for pest control raises questions about their potential environmental impact and safety for non-target organisms.

4. The study's findings need to be validated under field conditions to assess their practicality and effectiveness in real-world agricultural settings.

5. Line 81 could cite Lu, et al. (2020) Inbred varieties outperformed hybrid rice varieties under dense planting with reducing nitrogen. Scientific Reports10(1), 8769.

Author Response

Dear Prof. Sandra Wan

Thank you very much for your decision letter, concerning of our manuscript entitled “Metabolomic Profiling Reveals the Anti-herbivore Mechanism of Rice (Oryza sativa)” (ijms-2991558). According to the reviewers’ comments and suggestions, we have carefully revised our manuscript and our revisions are listed in the document designated as Response to Decision Letter. We provided the revised manuscript (ijms-2991558), the changes had been marked in red. Our revisions are listed as followings:

  • Responses to Reviewer 2:
  1. The molecular mechanism will be studied in the future.
  2. In this paper, only the effects of metabolites on the growth of Chilo suppressalis were determined, and the effects of active compounds on the growth and development of other insects will be determined in the future.
  3. All the bioactive compounds are secondary metabolites of rice, which are easy to degrade and have no effect on the environment at present. We will evaluate its safety again before it is popularized.
  4. The anti-pest activity of these compounds will be tested in the field in the future
  5. Line 81 had cited Lu, et al. (2020) Inbred varieties outperformed hybrid rice varieties under dense planting with reducing nitrogen. Scientific Reports, 10(1), 8769.

We hope that the revised manuscript will meet your magazine’s standard and is acceptable to be published in international journal of molecular sciences. Please feel free to contact me if you have any questions. We look forward to hearing from you regarding its disposition. Thank you again for your time and kind advice.

With best regards

Sincerely yours,

Cheng-Zhen Gu, Ph.D,

College of Life Science, Fujian Agriculture and Forestry University.
